# Prevalence of Undernutrition and Effect of Body Weight Loss on Survival among Pediatric Cancer Patients in Northeastern Hungary

**DOI:** 10.3390/ijerph18041478

**Published:** 2021-02-04

**Authors:** Orsolya Kadenczki, Attila Csaba Nagy, Csongor Kiss

**Affiliations:** 1Department of Pediatrics, University of Debrecen, 4032 Debrecen, Hungary; kisscs@med.unideb.hu; 2Department of Preventive Medicine, Faculty of Public Health, University of Debrecen, 4032 Debrecen, Hungary; nagy.attila@sph.unideb.hu

**Keywords:** nutrition, cancer, pediatric, survival, weight loss

## Abstract

Undernutrition is a prevalent condition in pediatric malignancy patients leading to unfavorable outcomes. The aim of this retrospective study was to determine the nutritional status and rate of undernutrition in 174 Hungarian pediatric patients with malignancies and the impact on 5-year survival based on anthropometric measurements. At the time of diagnosis, 5.0%, 4.6%, and 4.0% of patients were undernourished as determined by body weight (BW), weight-for-height (WFH), and body mass index (BMI) Z-score, respectively. The rate of undernutrition was 30.5% using ideal body weight percent (IBW%). Undernutrition at the time of diagnosis worsened the five-year overall survival only in solid tumor patients as defined by BMI Z-score and IBW%. Furthermore, 26.5% of patients became undernourished based on IBW% during the treatment period. Deterioration of nutritional status during treatment unfavorably influenced overall survival in both hematological and solid tumor subsets. Abnormal BW, WFH, and BMI Z-score were associated with poor prognosis in the hematologic group. The mortality risk was higher among hematologic patients with weight loss exceeding 20%. In conclusion, IBW% seems to be the most sensitive parameter to estimate undernutrition. Furthermore, BMI Z-score in both groups and severe weight loss in the hematological group may influence clinical outcome and play a role in prognosis assessment.

## 1. Introduction

Undernutrition is a big global health problem. Children with undernutrition are at major risk, especially if suffering from cancer and other chronic diseases [1]. Malnutrition, i.e., both undernutrition and overnutrition of children with malignancies were correlated with higher relapse and mortality rates and a higher risk for developing infections [2,3,4,5]. The etiology of cancer cachexia is multifactorial, and all of those factors may result in increased energy requirements, micro- and macronutrient needs, increased losses, or decreased intake of nutrients. Multifactorial aspects involve the disease itself and may have effects on therapy and on the inflammatory responses of the host. Insufficient growth and worsening organ functions (in particular immunocompromise) increase the risk of infections and, consequently, may increase treatment-related morbidity and mortality. Significant modifications in quantity and quality of dietary intake, changes in physical activity, and shifts in endocrine and metabolic functions may all contribute to an unfavorable alteration of energy balance in this population. Pharmacokinetics and pharmacodynamics of several anticancer agents can be altered by nutritional status. Undernutrition can influence the clearance of certain drugs, e.g., methotrexate clearance decreases in undernourished patients increasing toxicity. These complex interactions may explain how nutritional status may affect treatment outcomes of children with cancer, both in terms of morbidity and mortality [6,7].

The prevalence of undernutrition in children with cancer varies between 10% and 60%, depending on the criteria and assessment of undernutrition, types and stages of malignancy, the time of assessment during anti-tumor treatment, and the socioeconomic status of the examined patient subset [8,9]. Higher prevalence data of undernutrition have been published from low-GDP countries, where even most healthy children are undernourished [8,10]. In contrast, undernutrition is less frequent and oncological treatment outcome measures are better in high-GDP countries [8,9,11]. The nutritional status of Hungarian children with malignancies has not yet been determined. In Hungary, a country with a GDP lagging behind the most developed western European countries, the incidence of malnutrition in the general pediatric population is similar to the global average prevalence but it is higher than figures characterizing Germany and the Czech Republic [12,13].

Clinical outcome measures of children with malignancies in Hungary are approaching but do not reach those of wealthier, more industrialized countries [14,15,16]. Because the array of available drugs and anticancer and supportive treatment recommendations in Hungary are similar to those applied in western Europe and northern America, the slightly lower overall and event-free survival rates (OS, EFS, respectively) observed in our country may be influenced by suboptimal nutritional status as well.

Improper nutritional condition in this population has been most frequently screened by body mass index (BMI) [17,18]. However, BMI is less than optimal for the assessment of nutritional status in children with malignancies. BMI, which is calculated from height and weight, does not detect body composition. In children with malignancies, BMI may remain the same in course of treatment, while body fat-free mass may proportionally decrease in parallel with the increase in body fat mass [19]. Moreover, adequate, or optimal BMI percentiles may not reflect weight loss during anticancer therapy if the patient was overweight or obese at the time of diagnosis [19]. Different studies may not be comparable because cut-off points of impaired nutritional status may be different. For example, in some studies, BMI was considered abnormal below the –2 Z-score, whereas in some others, suboptimal BMI was considered below the 5th or the 10th centile values [20,21,22]. Therefore, BMI may be considered only one of the multiple parameters that detect undernutrition in children with malignancies.

Therefore, the aims of the present study were (i) to assess the prevalence and severity of undernutrition among children with malignancies in northeastern Hungary, (ii) to implement anthropometric parameters, such as ideal body weight % (IBW%) and body weight loss % (BWL%) in addition to BMI in order to develop tools for more reliable evaluation of undernutrition, and (iii) to correlate nutritional status of patients with survival rates.

## 2. Materials and Methods

### 2.1. Patients

Eligible patients for this retrospective, observational study were pediatric patients with cancer aged 1–18 years old, diagnosed between 1999 and 2009 and treated with chemotherapy in a tertiary care center for Pediatric Hematology-Oncology at the Department of Pediatrics, University of Debrecen. During this period, 218 children received active anticancer treatment according to current Hungarian Pediatric Oncology-Hematology Group guidelines. The management of children with cancer was strictly uniform in every Hungarian pediatric cancer treatment center [23]. Patients were followed for five years. Those 44 patients were excluded, who were lost to follow-up, or were transferred to another pediatric cancer treatment center for further treatment and follow-up. Finally, 174 patients were included in the study. Appendix A illustrates the flowchart of patient outcome and the composition of the main patient groups by diagnosis. The study was approved by the Scientific Research Ethical Committee of the Medical Research Council of Hungary (DE KK RKEB/IKEB No. 5623-2020) and was performed according to the 2008 Declaration of Helsinki. Written informed consent was obtained from legal guardians of participating patients.

### 2.2. Assessment of Nutritional Status

Body weight (BW; in kg) and height (in cm) of patients were recorded in their medical documentation from diagnosis until the end of anticancer treatment or death. BW, weight-for-height (WFH), and BMI (kg/m^2^) standard deviation scores (SDS; Z-score) were calculated based on the reference values of the Hungarian longitudinal child growth survey at the time of diagnosis, at the timepoint of the lowest BW during treatment and at the end of treatment [20,22]. Ideal BW percentage (IBW%; actual BW × 100/50th percentile WFH) values were calculated at the same time points. The percentage of weight loss (BWL%) was determined at the time point of the lowest BW during therapy (lost BW × 100/BW at diagnosis). Finally, undernutrition was defined as BW or WFH or BMI Z-score <–2.0, based on criteria of WHO or IBW% <90% or >10% BW loss (BWL%) during follow-up [17,21]. Patients were divided into three groups according to their BW loss—A: mildly (<20%); B: moderately (20–30%); and C: severely (>30%) undernourished.

### 2.3. Statistical Analysis

Normality was evaluated by the Shapiro–Wilk test. Pearson correlation was used to quantify the strength of association between two continuous variables. Multivariate logistic regression models were created for multivariate analyses. The five-year overall survival and event-free survival were calculated. The independent variables were analyzed for five-year OS and EFS for sex and age. Kaplan–Meier survival curves were estimated, and log-rank tests were used to compare them. The multivariable Cox proportional hazard model was used to calculate the hazard ratios of the OS and EFS. Intercooled Stata version 10 (StataCorpLLC, TX, USA) was used for the analysis and *p* < 0.05 was considered significant.

## 3. Results

### 3.1. Patient Characterization and Follow-Up

The 174 patients included 100 with hematological (57.5%) and 74 solid malignancies (42.5%). Patients were divided into these two main groups because of different disease characteristics and anticancer treatment protocols. The relatively low case numbers did not permit subgroup analysis of various specific tumor types. The median age at diagnosis was 7.34 years old (range: 1.09–17.06 years old). Altogether, 100 patients (57.5%) were males. Eventually, 137 patients (78.7%) were followed until the end of therapy, 2 hematologic patients relapsed during treatment and 15 died. There were two relapses among the solid tumor patients, 6 died during treatment and 12 had progressive disease without remission, who died before the end of therapy. Altogether, 33 patients died during therapy, and 36 died during the five-year follow-up. Characteristics of the 174 patients are shown in Table 1.

### 3.2. Nutritional Status of Patients at Baseline and during Follow-Up

Nutritional characteristics of the studied population are shown in Table 2. The BW, WFH, and BMI Z-scores did not differ significantly from Z = 0 at the time of diagnosis. IBW% was >90%. First, we compared BW, WFH, and BMI Z-scores at the three different time points (Table 2 and Table 3). Table 3 shows the level of significance of differences between the mean values of examined parameters. In general, patients lost BW during treatment, the median values significantly decreased, and the nutritional status of patients was worse at the second time point of the study compared to baseline (time of diagnosis). Until the end of treatment, BW increased, meaning values were higher or similar compared to baseline values.

The incidence of undernutrition defined by different anthropometric parameters described above is very variable, their sensitivity was significantly different. IBW% marked more patients undernourished than any other indices at any time point (Table 4.) At the time of diagnosis, 5.0%, 4.6%, and 4.0% of patients were undernourished as determined by BW, WFH, and BMI Z-scores, respectively. When applying IBW%, 30.5% of patients had IBW% <90%. This was observed in 26.0% of patients with hematologic malignancies and in 32.4% of those with solid tumors.

During treatment, patients lost BW and the rate of undernutrition increased. IBW% seemed to be the most sensitive tool to assess nutritional status. According to IBW%, 57.0% of patients were undernourished. When BWL% >10% was considered, 44.94% of patients had undernutrition. With respect to severity, 31.1% of all patients were mildly and 12.9% were moderately undernourished. Among patients with hematologic malignancies, 32.9% of children were mildly and 11.3% were moderately undernourished, while among solid tumor patients, 28.5% were mildly and 15.0% were moderately undernourished. We identified one single patient with non-Hodgkin lymphoma at diagnosis as severely undernourished as defined by BWL%.

By the end of treatment, the rate of undernutrition improved in the hematologic malignancy group as determined by IBW% (7.0%). Children with solid tumors remained undernourished as indicated by IBW% (36.2%).

### 3.3. Survival and Nutritional Status

Outcome and survival are highly affected by the nature of malignancy. According to our logistic regression analysis, the risk of mortality (odds ratio; OR) was 2.54-times higher among patients with solid tumors, compared to those with hematologic malignancies (*p* < 0.001, CI: 1.53–4.22).

At the time of diagnosis, favorable values of measured and calculated indices of nutritional status (BW, WFH, BMI Z-scores, and IBW%) had a protective effect—greater values were associated with less mortality risk of patients. The same association with calculated indices was observed at the end of treatment. Five-year OS and EFS, which were determined by Cox regression analysis, revealed that the lower the BW and calculated indices, the higher the risk of death, relapse, or secondary malignancy in patients (hazard ratio (HR): 3.36, 95% CI: 1.47–7.71, *p* = 0.004) (Figure 1.) Undernutrition at the time of diagnosis as defined by BMI Z-score (HR: 4.54, 95% CI: 1.48–13.97, *p* = 0.0081) and IBW% (HR: 2.71, 95% CI: 1.45–5.07, *p* = 0.002) significantly impaired five-year OS in patients with solid tumors.

Deterioration of nutritional status in course of treatment unfavorably influenced OS in both patient subsets. Abnormal BW (HR: 7.2, 95% CI: 2.59–20.49, *p* < 0.001), WFH (HR: 2.85, 95% CI: 1.1–7.38, *p* = 0.03) and BMI (HR: 5.98, 95% CI: 2.13–16.74, *p* < 0.001) were significantly associated with OS in the hematologic group. In patients with solid tumors, WFH (HR: 2.67, 95% CI: 1.3–5.49, *p* < 0.001) and BMI (HR: 2.47, 95% CI: 1.19–5.14, *p* = 0.015) had similar results. IBW% had significant effects only in the solid tumor group (HR: 3.79, 95% CI: 1.25–8.24, *p* < 0.001). BWL% impaired survival rates in all patient groups (HR: 4.13, 95% CI: 2.1–8.12, *p* < 0.001). The mortality risk was significantly higher among patients with hematological neoplasias whose BWL% exceeded 20% (HR: 6.87, 95% CI: 2.49–18.98, *p* < 0.001) (Figure 2). Moderate BWL% did not impair EFS among patients with solid tumors (HR: 1.49, 95% CI: 0.59–3.72, *p* = 0.39).

In the total study population, these indices had no effect on survival when assessed at the end of treatment as BW, and calculated indices of nutritional status improved by this time point. BW, WFH, and BMI status at the end of treatment had no effects on OS in the hematologic malignancy group (Appendix A). The patients were no longer undernourished at this time, and survival was not influenced by the end-therapy nutritional status. In contrast, BMI status had a pronounced effect on EFS in patients with solid tumors at all time points, in particular, at the end of treatment (HR: 8.4, 95% CI: 1.77–40.52, *p* < 0.001) (Figure 3). All patients with solid tumors who remained severely undernourished by the end of treatment, as defined by BMI, had progressive disease and succumbed to their diseases (Appendix A).

## 4. Discussion

The present study is the first in Hungary that assessed the nutritional status of pediatric cancer patients from the time of diagnosis until the end of treatment, and the effects of undernutrition on survival. Data for this observational retrospective clinical study were collected from a tertiary pediatric hematology-oncology care center in northeastern Hungary. Diagnostic and management guidance are uniform in all Hungarian pediatric cancer treatment centers and childhood cancer epidemiology figures of this region have been shown to represent well the whole country [24]. The take-home message of this investigation is that undernourishment at various time points of treatment of children, as characterized by different directly measured and calculated indices, had an untoward impact on disease outcome.

Nutritional status was defined by anthropometric parameters. In our study, while 4–5% of patients were found to be undernourished as determined by BW, WFH, and BMI Z-scores, 30% of patients were identified as undernourished by IBW%. Thus, IBW% identified a much higher percentage of patients as undernourished. Most frequently, however, BMI Z-scores were suggested for the screening of undernutrition and for the follow up of the nutritional status in patients with cancer [9,17,22]. Undernutrition is a common complication of children with malignant disorders—its prevalence varies from 5% to 60% [8,9,11]. Our results, based on the calculation of BMI Z-scores, tend to be similar to those reported from high-GDP countries. Publications from low-GDP countries present worse results at the time of diagnosis [8,10,11]. Cancer treatment exerted unfavorable effects on the nutritional status of our patients. Depending on the assessment tool used, 14–21% of the patients became undernourished. However, until the end of treatment, most of them succeeded in regaining weight. A similar tendency was reported by Brinksma et al. among pediatric cancer patients treated by active anticancer therapy [19]. In their study, however, changes during 3–12 months of treatment could not be related to gender, age, diagnosis, initial nutritional status, energy ingestion, and intensity of treatment. They observed a rapid increase of BMI and fat mass as determined by bioelectrical impedance analysis mainly in the first 3 months. These early increases in BMI could also be explained by intensive tube feeding and inadequate physical activity [19].

In the studies of Zimmermann et al. and Martin et al., 41–47% of children with malignancies became undernourished in course of anticancer management due to significant weight loss as defined by BMI ≤–2 SDS [25,26]. These authors explained undernutrition by age (>10 years old), type of malignancy, and the emetic effects of anticancer drugs. In our study, it was not possible to analyze different types of diagnosis or different age subsets because of the low patient numbers. Yet, we found undernourishment in course of anticancer chemotherapy in 11.1% and 26.8% of our patients with hematologic and solid malignancies, respectively.

The sensitivity of anthropometric parameters in our study was similar, except IBW%. For the calculation of IBW%, we used the 50th percentile WFH of healthy children. If IBW% was <90%, the patient was undernourished. In our population, more children were undernourished when determined by IBW% compared to those assessed by BW, WFH, or BMI Z-scores. At the time of diagnosis and during treatment, 30.5% and 57% of children were undernourished as defined by IBW%. There were no significant differences between patients with solid or hematologic malignancies. We obtained a similar figure (44.94%) using BWL%. WFH and BMI Z-scores are the recommended indices for undernourishment among cancer patients. Similar to Shah et al., we found these two parameters less sensitive for screening because, in contrast to IBW% or BWL%, WFH and BMI identified only some of the undernourished patients [11]. We consider that relatively physiological WFH and BMI scores during treatment and at the end of treatment can be influenced by faltering growth. The age of patients was not taken into consideration when calculating IBW%, and only the required WFH was considered [19]. This may be the reason why we obtained different results when calculating BW and BMI Z-scores. Regarding BWL%, the new BW is compared to the baseline BW of the same patient. This calculation may be misleading in those patients who lost weight before the disease onset or in those chronically undernourished. Oncologists take these factors into consideration during the screening of adult patients, and they ask patients about weight loss during the previous 3 months [27,28,29]. In pediatric oncology centers, this issue may not draw enough attention [30]. This may be the reason why, mainly in solid tumor patients with BW loss already present before diagnosis, results do not reflect their real nutritional status. In addition, their survival rates based on anthropometric data may look better than they really are.

With respect to the possible role of undernourishment in survival, it is still uncertain which anthropometric parameter would be the most sensitive and the most useful in these patient cohorts. In children with malignancies, survival may be mostly determined by BWL%. Among our patients with acute lymphoblastic leukemia (ALL) having optimal nutritional status at the time of diagnosis, mortality risk was increased if BWL% exceeded 20–30% in course of treatment. The survival of children with solid tumors was worse than that of children with hematological malignancies. We have no direct explanation for this observation. We cannot determine to what extent did chronic undernourishment contribute to the eventual death of individual patients. Finally, there are no hard endpoint longitudinal studies proving that proper nutritional intervention in patients with solid tumors would improve overall survival.

In the 1990s, undernutrition was an independent risk factor of mortality in ALL patients in developing countries. Today, mainly in high-risk ALL, infections and treatment-related mortality are the most important determinants of survival. Undernutrition may have less importance in this respect [10,31,32]. Nowadays, overweight and obesity rather than undernutrition seem to worsen OS and EFS survival in industrialized countries [4,33]. In our study, there were nine high-risk cases among the 49 ALL patients, and three of them died. All nine high-risk patients were in remission but undernourished in course of therapy. In two patients, IBW% was <80% and BWL% was almost 20%. Undernutrition was not so common in other ALL cohorts [28]. As Brinksma et al. reported, in patients with ALL, the prevalence of undernutrition at diagnosis decreased from 5–10% at baseline to 0–5% by the end of treatment [34]. We cannot fully explain why undernutrition was more common among our patients with ALL and other hematologic malignancies.

Our study has strengths and weaknesses. This is the first study assessing the nutritional status of children with malignancies in Hungary. We suggested that BWL% over other calculated anthropometric parameters may be the most accurate indicator of nutritional status in course of the treatment and follow-up of such patients. Other anthropometric markers, such as BMI, WFH, and BW Z-scores, and IBW% may still show normal nutritional status in otherwise undernourished children. Other groups also found that determining ideal body weight and BWL% were the most sensitive parameters for controlling the nutritional status of oncologic pediatric patients [19,30].

Limitations of our study include the relatively low number of patients, which did not permit subgroup analyses. Therefore, we could not analyze separately the impact of different types of cancer, different treatment protocols, different stages of the disease, and socioeconomic status, which may all exert an effect on the nutritional status and, together with the nutritional status, influence treatment outcome of these children. Prospective multi-centric studies investigating a larger number of children with cancer are required to unravel the effect of nutritional status of children receiving contemporary anticancer treatments more in detail. Moreover, we did not investigate potential weight loss before diagnosis resulted from the neoplastic disease itself because, in most cases, we did not have exact data characterizing the nutritional status of children prior to diagnosis.

The consequences of malnutrition and inadequate support or nutrition (enteral and parenteral) are important aspects of supportive care, which unfortunately have been too frequently neglected due to the focus on anticancer treatment. These findings, together with similar observations from more Hungarian pediatric cancer treatment centers, prompted us to prepare guidance for nutritional support of children during anticancer management. Results of the introduction of this guidance have not been reflected, however, in this retrospective cohort [9].

## 5. Conclusions

Our results verified the unfavorable effect of undernutrition on survival in children with cancer. Recognizing undernutrition as early as possible should be desirable using the simplest method. Following the change of body weight and defining deterioration of body weight in percentage seems to be a reasonable method when lacking equipment for precise measuring of body composition (e.g., bioelectrical impedance analysis). Because undernourishment is closely related to survival, our data also underscore the importance of close monitoring of the nutritional status in children with malignancies.

## Figures and Tables

**Figure 1 ijerph-18-01478-f001:**
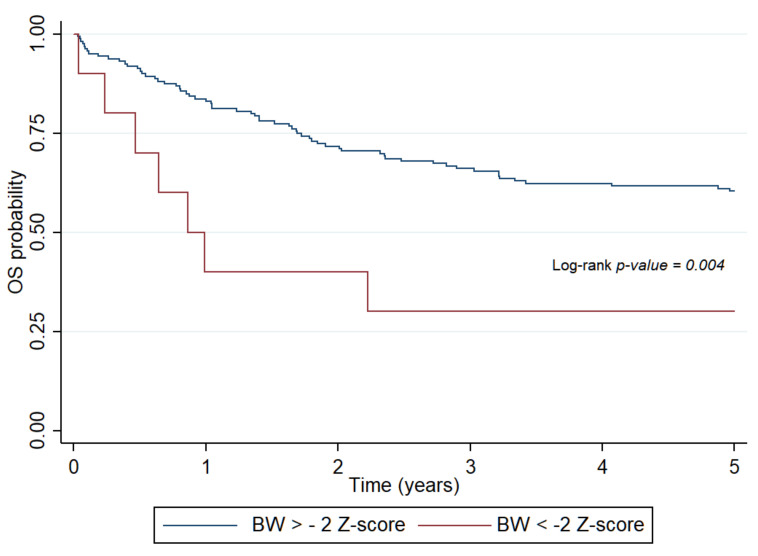
Probability of overall survival for patients with pediatric hematologic malignancies determined by body weight Z-score at diagnosis. OS: overall survival, BW: body weight.

**Figure 2 ijerph-18-01478-f002:**
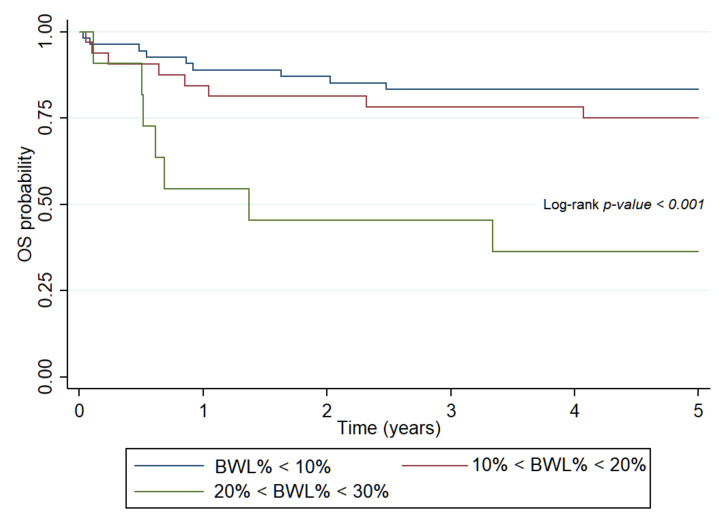
Probability of overall survival for patients with hematologic malignancies determined by body weight loss percent in course of treatment. OS: overall survival, BWL%: body weight loss percent.

**Figure 3 ijerph-18-01478-f003:**
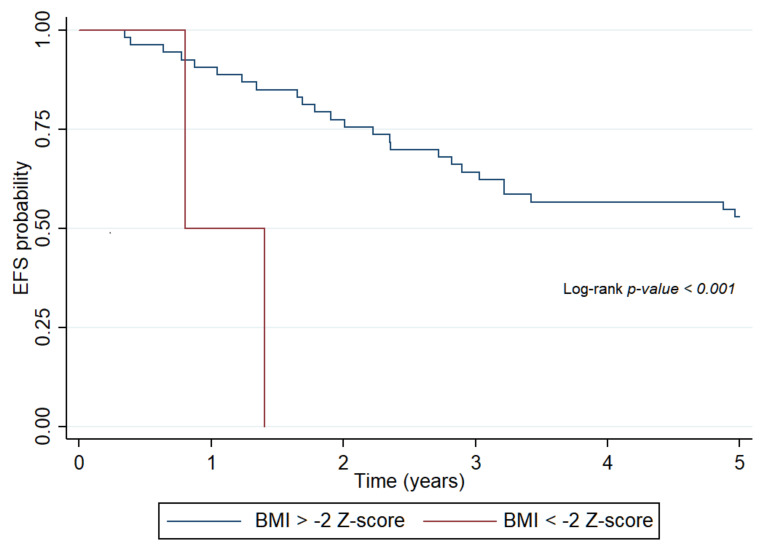
Probability of event-free survival of patients with solid tumors determined by body mass index Z-score at the end of treatment. EFS: event-free survival, BMI: body mass index.

**Table 1 ijerph-18-01478-t001:** Patients demographic data.

Tumor Subtypes	Number	Mean Age		Gender		Outcome	
				Male/Female		Alive/Relapsed	Death
		(Years ±SD)	(Min–Max)	*N*	%	*N*	*N*
Total	174	7.34 ± 5.10	(1.01–16.79)	100/74	57.47/42.53	106	69
Onco-hematologic malignancy	100	8.04 ± 4.84	(1.16–16.79)	57/43	57.00/43	73	26
Solid tumor	74	6.40 ± 5.32	(1.01–17.07)	43/31	58.11/41.89	33	43

*N*: number of patients; SD: standard deviation.

**Table 2 ijerph-18-01478-t002:** Changes in mean values of body weight, weight-for-height, body mass index Z-scores, and ideal body weight percentage at three follow-up time points.

Follow-Up Points									
Patient Group	At Diagnosis			Nadir during Therapy			End of Therapy		
	Mean	SD	Min–Max	Mean	SD	Min–Max	Mean	SD	Min–Max
BW									
total	−0.18	1.29	−2.76–5.31	−0.87	1.26	−3.92–4.81	−0.18	1.33	−3.51–6.01
H	−0.068	1.2	−0.31–0.18	−0.71	1.27	−0.96–−0.46	0.16	1.29	−0.11–0.44
S	−0.33	1.32	−2.75–5.31	−1.1	1.21	−3.25–4.81	−0.70	1.22	−3.51–4.46
WFH									
total	−0.29	1.19	−3.0–3.48	−1.09	1.15	−4.44–2.99	0.25	1.35	−3.24–5.77
H	−0.20	1.1	−0.42–0.12	−0.99	1.11	−1.21–−0.77	0.4	1.34	0.11–0.69
S	−0.41	1.3	−3.0–3.46	−1.23	−3.63	−3.63–4.81	−0.53	1.18	−3.24–3.28
BMI									
total	−0.17	1.21	−2.66–4.84	−0.97	1.18	−2.66–4.84	0.085	1.3	−2.94–5.08
H	−0.07	1.15	−0.3–0.15	−0.86	1.13	−1.09–−0.63	0.41	1.26	0.14–0.68
S	−0.3	1.28	−2.66–4.84	−1.11	1.24	−3.33–4.27	−0.4	1.21	−2.94–4.2
IBW%									
total	99.15	15.6	62.96–161.53	88.87	14.1	56.6–153.84	103.12	17.5	69.41–166.46
H	100.46	14.79	97.52–103.39	90.19	12.96	87.6–92.78	108	17.42	104.26–111.73
S	97.39	16.69	62.96–161.53	87.16	15.46	83.5–90.82	95.88	15.05	91.92–99.84

BW: body weight, WFH: weight-for-height, BMI: body mass index, IBW%: ideal body weight percent, SD: standard deviation, H: hematologic malignancy, S: solid tumor.

**Table 3 ijerph-18-01478-t003:** Compering mean values of anthropometric data at the three examined timepoint.

Anthropometric Parameter	Tumor Subtypes	During Therapy to Diagnosis	During Therapy to End of Treatment	Diagnosis to End of Treatment
		*p*	*p*	*p*
BW	Total	<0.001	<0.001	0.970
	H	<0.001	<0.001	0.215
	S	<0.001	0.006	0.200
WFH	Total	<0.001	<0.001	<0.001
	H	<0.001	<0.001	<0.001
	S	<0.001	<0.001	0.590
BMI	Total	<0.001	<0.001	0.070
	H	<0.001	0.007	0.007
	S	<0.001	0.001	0.650
IBW%	Total	<0.001	<0.001	0.033
	H	<0.001	<0.001	0.002
	S	<0.001	0.002	0.590

*p*: *p*-value (level of significance), H: hematologic malignancy, S: solid tumor, BW: body weight, BMI: body mass index, WFH: weight-for-height, IBW%: ideal body weight percent.

**Table 4 ijerph-18-01478-t004:** Rate of incidence of undernutrition based on different indices.

Timepoint	BW	BW	BW	WFH	WFH	WFH	BMI	BMI	BMI	IBW%	IBW%	IBW%	WL%	WL%	WL%
	Total	H	S	Total	H	S	Total	H	S	Total	H	S	Total	H	S
	(%)			(%)			(%)			(%)			(%)		
At diagnosis	5.75	5	6.76	4.62	4.04	5.41	4.02	3	5.11	30.45	26	32.4	NA	NA	NA
During treatment	14.11	12.12	16.9	20.58	15.15	28.1	17.6	11.11	26.7	57	55.55	59.1	44.94	43.43	43.5
End of treatment	4.1	0	10.34	3.4	0	8.6	1.38	0	3.44	18.75	6.98	36.2	NA	NA	NA

BW: body weight, WFH: weight-for-height, BMI: body mass index, IBW%: ideal body weight percent, WL%: weight loss percent, H: hematologic malignancy, S: solid tumor, NA: not applicable.

## Data Availability

Datasets analyzed during the study represent patient’s data available in their medical documentation and the electronic patients’ database (MedSolution) of the University of Debrecen for authorized personnel. Anonymized compilation of dataset analyzed in the study is also available from the first author of this article upon request.

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
