# Peer review of "Prevalence of Undernutrition and Effect of Body Weight Loss on Survival among Pediatric Cancer Patients in Northeastern Hungary"

_ijerph, 2021, doi:10.3390/ijerph18041478_

Round 1

Reviewer 1 Report

The authors present a paper : " Prevalence of undernutrition and effect of body weight loss on survival among pediatric cancer patients in North-Eastern Hungary" interesting as content but with significant  limitations:

  1. relatively low number of patients (as reported by Authors)
  2. circumscribed to a part of a Country
  3. unconvincing conclusions
  4. lack of effective suggestions on how to prevent the clinical conditions
  5. a clear explanation why deterioration of nutritional status during treatment unfavorably influences overall survival in both hematological and solid tumor subsets.
  6. what is the take-home message from this study?

Reviewer 2 Report

I think the main focus of this study might need to be re-defined: what is the causal relationship between malnutrition and pediatric cancer. The authors studied one direction: how malnutrition affects cancer progression but not the other direction: how cancer affects malnutrition.  This need some causal inference study here, which I believe need some major revisions.

Some minor comments:

1) Table 2 need frequency report. Just P-values are not enough. Need to move supplemental table 1 and table 2 to main text.

2)Line 156, the sentence need to be re-written with grammatical mistakes.

3) Need to report P-values for K-M curves in Figure 1, 2 and 3. 

Reviewer 3 Report

Authors worte an important paper on big issue. Malnutrition represent a big public health /global health problem 

Only some minor suggestions:

  1. Introduciton. Undernutrition is a big gloabal health problem. Children with undernutrion can be defined " Children at risk" because they are most vulnerable and their health and social oucome are a risk. (see and cite this experience from Mozambique: The At Risk Child Clinic (ARCC): 3 Years of Health Activities in Support of the Most Vulnerable Children in Beira, Mozambique" Int. J. Environ. Res. Public Health 15, no. 7: 1350). Add data on global burden of Malnutrition
  2. Methods and results: are clear
  3. Discussion:discuss better also the correlation with infectious dieseases and which public health solutions are viable.

Round 2

Reviewer 1 Report

The Authors responded carefully and clearly to my requests and I am satisfied because they made the manuscript more complete .